# Visualizing changes to US federal environmental agency websites, 2016–2020

**Eric Nost**[1,2]*, **Gretchen Gehrke**[2], **Grace Poudrier**[2,3,4], **Aaron Lemelin**[2], **Marcy Beck**[2], **Sara Wylie**[2,4,5], **on behalf of the Environmental Data & Governance Initiative**[2¶]

**1** Department of Geography, Environment and Geomatics, University of Guelph, Guelph, Ontario, Canada, **2** Environmental Data & Governance Initiative, United States of America, **3** Department of Sociology and Anthropology, Northeastern University, Boston, Massachusetts, United States of America, **4** Social Science Environmental Health Research Institute, Northeastern University, Boston, Massachusetts, United States of America, **5** Department of Sociology/Anthropology and Health Sciences, Northeastern University, Boston, Massachusetts, United States of America

¶ Membership in the Environmental Data & Governance Initiative is available at https://envirodatagov.org/about/members-2/
* enost@uoguelph.ca

**Data Availability Statement:** The data underlying the results presented in the study are available from the following Github repository: https://github.com/edgi-govdata-archiving/web_monitoring_research.

## Abstract

Websites have become the primary means by which the US federal government communicates about its operations and presents information for public consumption. However, the alteration or removal of critical information from these sites is often entirely legal and done without the public's awareness. Relative to paper records, websites enable governments to shape public understanding in quick, scalable, and permissible ways. During the Trump administration, website changes indicative of climate denial prompted civil society organizations to develop tools for tracking online government information sources. We in the Environmental Data & Governance Initiative (EDGI) illustrate how five data visualization techniques can be used to document and analyze changes to government websites. We examine a large sample of websites of US federal environmental agencies and show that between 2016 and 2020: 1) the use of the term "climate change" decreased by an estimated 38%; 2) access to as much as 20% of the Environmental Protection Agency's website was removed; 3) changes were made more to Cabinet agencies' websites and to highly visible pages. In formulating ways to visualize and assess the alteration of websites, our study lays important groundwork for both systematically tracking changes and holding officials more accountable for their web practices. Our techniques enable researchers and watchdog groups alike to operate at the scale necessary to understand the breadth of impact an administration can have on the online face of government.

## Introduction

As demonstrated by the social media-based manipulation of public knowledge during the COVID-19 pandemic [1], new ways are needed to track and contextualize efforts to influence public perception. This paper focuses on techniques for visualizing changes to government

**Funding:** EDGI is sincerely thankful for the support of its funders, including the David and Lucile Packard Foundation and the Doris Duke Charitable Foundation. Grace Poudrier also gratefully acknowledges funding from the National Institute of Environmental Health Sciences of the National Institutes of Health (T32ES023679). The funders had no role in study design, data collection and analysis, decision to publish, or preparation of the manuscript.

**Competing interests:** The authors have declared that no competing interests exist.

websites. We emphasize the topic of climate change, based on the well-documented and systematic ways distrust and confusion have been generated around it [2].

On October 17th, 2018, the US Environmental Protection Agency (EPA) made a small but illustrative change to one single page on its website. On a "Help finding information" page (https://www.epa.gov/sites/production/files/signpost/cc.html), the agency removed language that suggested it was updating an entire set of pages related to climate change. The help page had stated, "Thank you for your interest in this topic. We are currently updating our website to reflect EPA's priorities under the leadership of President Trump and Administrator Pruitt" [3]. In removing this line, the EPA revealed it was no longer updating—if it ever had been— the epa.gov/climatechange directory, which had already been withdrawn from public access for over a year.

The removal of epa.gov/climatechange illustrates the value of web monitoring. The directory was a vital resource for understanding climate change alongside federal climate research and mitigation and adaptation actions. Its termination signaled an important shift in ongoing efforts to influence public understanding of climate science: the spreading of climate denial through the federal agency dedicated to environmental protection.

Our web monitoring, part of the Environmental Data & Governance Initiative (EDGI), began as a page by page analysis of changes like this. This did not give us much of a comprehensive view, so we developed the techniques described in this paper for analyzing webpage changes systematically across federal agencies and surfacing how the Trump administration reframed public information around contentious issues such as climate change. Between Trump's election in November 2016 and his inauguration in January 2017, civil society groups had prepared for significant removals of environmental and public health data from public access. Organizations like ours partnered with others to "rescue" datasets presumed to be endangered [4, 5]. We took seriously Trump's overt denial of climate change and stated intention to reduce environmental protection to just "a little bit." [6] In the end, watchdog groups mostly did not observe outright data destruction, but an array of occurrences—from one-word changes to removals of entire pages and directories like epa.gov/climatechange—in which federal agencies altered or withdrew web information that contextualized data, the work of scientists, and government regulations [7].

Websites represent the most important way people learn about environmental issues from the US federal government, a seemingly authoritative source. As early as 2003, more than 50% of Americans and 75% of American internet users reported using government websites and web services [8]. By 2020, US federal agency websites received millions of visitors every year, and were linked to by myriad sources, including 25,000 linking to www.epa.gov [9]. The Office of Management and Budget recognizes that the web is "*the primary means* by which the public receives information from and interacts with the Federal Government" ([10], emphasis added).

Agency websites are key sources of information about environmental regulations in particular. They play a critical part in one of the most democratic processes built into federal environmental decision-making: public comments [11, 12]. For example, the Trump EPA removed its Clean Power Plan and Clean Water Rule websites well in advance of formal notification that it intended to repeal the rules, and more than two years before either was officially repealed. The entire Clean Power Plan website was redirected to a single webpage on "energy independence" [13], and the entire Clean Water Rule website was redirected to a small site strictly focused on the definition of Waters of the United States [14]. These changes left the public without federally-provided information about either of these landmark rules before, during, and after the period in which people could formally voice objections to their repeal. Website removals and modifications therefore do not just *reflect* an administration's stance on

issues such as climate change and public health but play an active part in advancing its agenda by framing what information is accessible to public discourse.

In this paper, we describe our application of five data visualization techniques to changes to US federal environmental agencies' websites between 2016 and 2020. First, we characterize existing research on how scientific and policy matters–particularly with respect to climate change—are communicated. We then review how federal records are governed and contextualize the permissibility of website changes by describing more progressive legal frameworks in other countries around the world.

Second, we introduce our five data visualization techniques and their application to federal websites. The first focuses on documenting whether links to climate- and environment-related pages have been removed and, as a result, if these pages are now less accessible. The rest illustrate changes to the use of key terms like "climate change": overall, in relation to one another, by agency, and by location in the agency's website structure (on highly visible landing pages or pages deeper in the weeds of a subject). Each offers a unique window into how public information changed from 2016 to 2020.

Finally, we discuss the value and limitations of our techniques. While edits to websites are to be expected–whether to reflect the latest science, indicate an update to policy, announce a new program, or correct a typo—the true impact of these changes can only be interpreted across many pages. Their analysis must be mostly automated, quantitative, summarizable, and visually interpretable. Each of our techniques has constraints on how much information it can glean, largely driven by our approach to compare pages with archives available for *both* 2016 and 2020—what we call a "paired-page" sample. Yet each can readily be applied to documenting, analyzing, and visualizing changes at the scale of the federal web presence. We generate critical new findings about the evolution of US federal websites between 2016 and 2020, with dramatic reductions in information about climate change—especially on highly visible and Cabinet-level agency webpages—and demonstrable shifts in rhetoric. Our results underscore the need for accountability around the provision of accurate and relevant information amidst propaganda—misinformation, censorship, and gaslighting—by state and corporate entities [15].

## Context and framework

We bring together bodies of knowledge on climate change communication, e-government, and the legal history of (web) records-keeping in order to contextualize website changes: why federal agencies use the web, why web changes occur, and why they matter. This frames our use of content and network analysis to count keywords such as "climate change" on webpages and the links between them. Since websites are the primary means by which the federal government informs the world about its actions and perspective on key topics, the removal of language—from deleting specific words to entire pages—can shape public understanding of environmental issues and reveal agencies' approaches to them.

Due in large part to a decades-long misinformation campaign coordinated by fossil fuel interests [2, 16–20]—and the ideological wedges that have resulted—the language used in everyday discourse, media, and politics to describe climate change mechanisms, impacts, and mitigation strategies has polarizing connotations [21–25]. For example, consider the distinctions between "hydraulic fracturing" (as used by some regulators), "fracking" (used by some environmental groups), and the umbrella terms "horizontal drilling" or "unconventional gas extraction" (used widely by industry). The term "climate change" itself and associated keywords, such as its colloquial precursor "global warming", have also become polarized over time [26, 27].

Historically, the climate denial movement focused on undermining prominent scientists, disputing climate models, disaggregating climate change from anthropogenic sources, and influencing policy-makers with cherry-picked data that downplayed climate changes' severity [2]. With the election of Trump, these tactics found new purchase *within* federal agencies. While agencies such as NASA and NOAA have proved vital in the development of climate change science, their role makes them and their websites targets for those seeking to sow doubt on climate change, be it think tanks such as the Heartland Institute [28], or their allies in government itself [29]. The manipulation of public understanding is, of course, nothing new to governments. As the author of *Propaganda* Edwards Bernays wrote in 1928: "the conscious and intelligent manipulation of the organized habits and opinions of the masses is an important element in democratic society." [30]

Climate change science has long been communicated to the public at least in part through government websites (epa.gov/climatechange was created during the Clinton administration). The information sciences literature on "e-government" theorizes that web technologies mediate the relationship between citizens and government [31–34]. In the late 1990s and early 2000s, scholars began evaluating the extent to which public information and service provision through the internet could enhance and expand democracy [35–37]. Against industry influence on knowledge production and dissemination, researchers speculated that social media, open data platforms, blogs, and other new media might lead to an increase in openness. While many governments embraced the web as a way to rebuild trust with citizens [38], the full democratic potential of e-government has not been actualized [36]. Federal government websites are often viewed as some of the most reputable sources of information (e.g. for health: [39], but public trust in them and adherence to messages they promote are variable [40], and can be related to the consistency of access to information [41]. Research suggests that a user's existing propensity to trust influences their use of e-government services, rather than the reverse—that e-government increases trust [42]. There has been opposition to government restrictions of information [41, 43], and a recognition that access restrictions typically have negative impacts on society [44].

US government agencies utilize digital media, including websites, in limited and problematic ways. However, many website changes, and the use of the web to advance policy agendas, are to be expected if not always appropriate. For instance, the Bush administration removed reams of supposedly sensitive information from agency websites after the September 11, 2001 terrorist attacks, leading to stark criticism [45]. While setting a historical precedent, these removals were not targeted towards specific policies but to types of information that could be used with nefarious intent. Basic resources, such as epa.gov/climatechange were retained. Later, the Obama administration pushed for open government policies, including public access to agency data [46]. However, it also controversially (and illegally) used social media to lobby for its 2015 Clean Water Rule [47, 48].

These examples and others show that website changes may be made to reflect new political priorities, the evolution of understanding on a scientific topic, to add coverage of emerging or emergency issues, or even as part of routine quality assurance processes. For example, in 2017, the National Park Service removed 92 national parks' Climate Action Plans from its website [49]. There was alarm amongst the scientific and NGO community, but the documents were eventually restored after they were updated to ensure compliance with accessibility requirements [50]. Website changes can also involve censorship, and may stoke polarization; they can be for societal benefit or infringements on democratic dialogue. For instance, the Trump administration limited access to information about health insurance enrollment, effectively impeding an unfavored policy, and also altered phrasing across entire web domains to refer to migrants using the term "alien," favoring xenophobic perspectives [51, 52]. Some media

commentators have taken such website changes as reflections of partisan politics at the highest levels of government (e.g. in [53]), as occurred in the Bush administration's web-scrubbing [41]. However, our own research and others' suggest that web changes are not always directly ordered by appointed politicians; often, they are a more complicated product of political climate, sometimes involving forms of self-censorship (see also [54–56]).

While websites are the primary means through which the public receives information from the government, they are not necessarily intentionally designed to facilitate learning outcomes. Federal websites are also not designed to provide other government personnel—such as congressional staffers and state and local officials—with access to information that can inform policies and practices. Content on federal websites is instead often more oriented towards industry than public audiences—promoting the use of technologies that run counter to prevailing public concerns about environmental issues, for example [57]. This industry-orientation is an issue when it conflicts with science and public health needs. Web changes are impactful in socially uneven ways. For instance, while the EPA created an official archive of its English-language climate change pages, it did not do so for the Spanish-language versions [58], even further narrowing access to these resources along an axis of social inequality.

All this underscores the continuing need to evaluate government website design and content [31, 32, 59–61]. Many such evaluations have been agency- or website-specific, intimately characterizing web content and the practices by which it is altered and maintained. Extensive, large-scale, replicable and visualizable techniques can complement intensive website analysis in e-government research.

In many cases, this will require access to records of websites as they appeared in the past. Several laws forbid the destruction or alteration of paper records, but these have rarely been updated to apply to digital records (i.e. web pages, databases, and "born digital" public information) [62]. Digital public information on climate change—including websites and databases of scientific information hosted under the federal.gov domain—is unusually vulnerable to politicization, alteration, and destruction relative to both US paper records as well as digital public information outside the US.

While governments have always preserved records in some form, the contemporary legal structure surrounding them dates to the mid-twentieth century. The foundation of US public records law, the US Federal Records Act of 1950 (FRA), requires federal agencies to create, maintain, and preserve public copies of all materials they create (44 U.S.C. § 3101). Although a 2014 effort to modernize the FRA was signed into law under the Obama Administration (Presidential and Federal Records Act Amendments of 2014), it did not establish explicit legal equivalence between electronic records and paper records; consequently, the removal of electronic records or links from.gov websites is not considered a form of public records destruction under the current FRA [62]. While the chief agency tasked with the management and oversight of federal records programs (the National Archives & Records Administration, NARA) has published guidelines for the management of federal agency webpages, it has no enforcement mechanism and by extension does not stipulate a penalty for altering or removing public records from the web [62, 63]. Legislative preservation requirements for federal agencies remain generally limited by a definition of "records" that does not encompass websites and are subject to interpretation by NARA archivists [64]. Furthermore, born-digital publications—content that is originally created in digital form rather than paper records that have been digitized—are exempt from US legal deposit laws, which require copies of all print works registered under US copyright to be sent to the Library of Congress [65].

Outside the US, several countries have established comparatively savvy legal frameworks to govern and preserve public digital information. Iceland was the first country to legally mandate web harvesting and has been performing "whole domain" crawls and archiving of

government webpages (.is domains) since 2004. New Zealand's legal deposit laws were extended to cover all born-digital materials, including government websites, in 2006 [66]. The United Kingdom (UK) similarly amended its legal deposit laws in 2013 to require that everything published on the UK web be deposited into the British Library; periodic domain-scale crawls of UK government websites are also legally mandated [67]. France utilizes an open source crawler-bot program to automatically archive all pages registered under the.fr domain on an annual basis to comply with legal deposit laws and the European Union established an official EU Web Archive (EUWA) in 2013 to combat the ephemerality of web-based information. All content hosted on the Europa.eu domain and subdomains are crawled and archived by the Publications Office of the European Union four times per year, and all archived content is made publicly available and searchable online [68].

There is currently no US equivalent to these legally-mandated practices. With no meaningful policies in the US for preserving information on federal agency websites, the not-for-profit Internet Archive and its Wayback Machine have functioned as the most comprehensive forms of archiving online government content. The Wayback Machine records "snapshots" of pages across the web at varying intervals, and has contributed to elucidating government, corporate, and media web practices and even solving terrorism cases [65]. The End of Term Web Archive is a collaborative project between the Library of Congress, US Government Publishing Office, Internet Archive, and several university libraries and has been preserving select US government websites at the end of presidential administrations since 2008. Publicly accessible copies of this archive are available through the Internet Archive for 2008, as well as successive years of presidential transition archives created in 2012 and 2016 [69]. A similar, albeit smaller, web transparency project is the University of North Texas's "CyberCemetery," an archive of defunct government agency websites that is maintained in partnership with the Government Publishing Office (GPO) and NARA. The Cemetery is not comprehensive and includes an FAQ section with the following plea to federal employees: "If your agency or commission is closing, please [contact us to] make sure your website's valuable information is preserved!" [70]

Web-based environmental information may make governments' decisions more capturable, because of the digital traces archives can record. But simply documenting webpages as they appeared in the past is limited; transparency does not automatically translate into accountability [71–73]. Our website monitoring is praxis for both documenting and analyzing website changes—a first step to holding government agencies to account [74]. Anecdotal evidence suggests that this practice can prove effective in mitigating further removals; an EPA staffer told a reporter that "No one is willing to touch the website because everyone's afraid of the news stories that say, 'EPA changed this´´´ [75].

## Materials and methods

In the rest of this paper, we show how data visualization techniques can help to characterize changes to US federal websites [76–79]. As EDGI cohered in early 2017, our Website Monitoring Team started building a database of webpages to monitor. At first, a small team of internal domain experts identified "seed pages" related to environment, climate, and energy, and then the public was invited to provide others. By 2018, EDGI's Website Monitoring Team had built a database of approximately 40,000 URLs (see Supplemental Materials A in S1 File), selecting these seed pages and crawling URLs from there for up to 50 steps, or until a URL limit had been reached. Each of these pages has been saved in the Internet Archive's Wayback Machine. The 40,000 URLs includes webpages from 13 federal agencies, though the majority are from

**Table 1. List of US federal agencies with environmental mandates, and whether they develop and enforce regulations, have a research mission, and/or are Cabinet-level.** We considered agencies within Cabinet departments (e.g. the Bureau of Land Management within the Department of the Interior) to be non-Cabinet agencies, whereas the departments themselves are Cabinet-level. We also considered the White House to be Cabinet-level.

| Agency | Environmental Regulations? | Research Mission? | Cabinet-level? |
|---|---|---|---|
| Bureau of Indian Affairs (BIA) | No | No | No |
| Bureau of Land Management (BLM) | Yes | No | No |
| Bureau of Ocean Energy Management (BOEM) | No | No | No |
| Center for Disease Control and Prevention (CDC) | No | Yes | No |
| General Services Administration (GSA) | No | No | No |
| Department of Energy (DOE) | Yes | No | Yes |
| Department of the Interior (DOI) | Yes | No | Yes |
| Department of Justice (DOJ) | No | No | Yes |
| Department of Transportation (DOT) | No | No | Yes |
| Energy Information Administration (EIA) | No | Yes | No |
| Environmental Protection Agency (EPA) | Yes | No | Yes |
| Office of the Federal Register | No | No | No |
| Federal Emergency Management Administration (FEMA) | Yes | No | No |
| Federal Energy Regulatory Commission (FERC) | Yes | No | No |
| US Forest Service (USFS) | Yes | No | No |
| Fish and Wildlife Service (USFWS) | Yes | No | No |
| Government Accountability Office (GAO) | No | Yes | No |
| Global Change Research Program (GCRP) | No | Yes | No |
| Department of Health and Human Services (DHS) | No | No | Yes |
| National Aeronautics and Space Administration (NASA) | No | Yes | No |
| National Institutes of Health (NIH) | No | Yes | No |
| National Oceanic and Atmospheric Administration (NOAA) | Yes | Yes | No |
| National Park Service (NPS) | No | No | No |
| National Science Foundation (NSF) | No | Yes | No |
| Occupational Safety and Health Administration (OSHA) | Yes | No | No |
| Office of Surface Mining Reclamation and Enforcement | Yes | No | No |
| US Department of Agriculture (USDA) | Yes | No | Yes |
| US Geological Survey (USGS) | No | Yes | No |
| White House | No | No | Yes |

EPA, DOE, NASA, and NOAA (see Table 1 for a description of these acronyms). We achieved broad but not full coverage of federal environmental websites.

Our team of analysts monitored specific subsets of our database on a weekly basis, making it possible to review changes over the course of the previous week. We always realized that week to-week web monitoring would be limited. These constraints include volunteer capacity as well as the ability to synthesize across previous changes. Analysts focused on week-to-week changes to their assigned pages, with no one specifically dedicated to researching thematic-based reports. Documenting changes in real-time was necessary to hold Trump agencies accountable for their censorship, neglect and omission of science, and industry-orientation at odds with protection of human and environmental health.

Yet we also needed to be able to articulate the breadth and pattern of changes more systematically. To address the constraints of a pointillist approach to web monitoring, we first undertook a content analysis—a historical and comparative approach to examine shifts in the use of keywords such as "climate change." Content analysis is a widely used approach in the social sciences that helps researchers understand the prevalence of certain ideas based on their

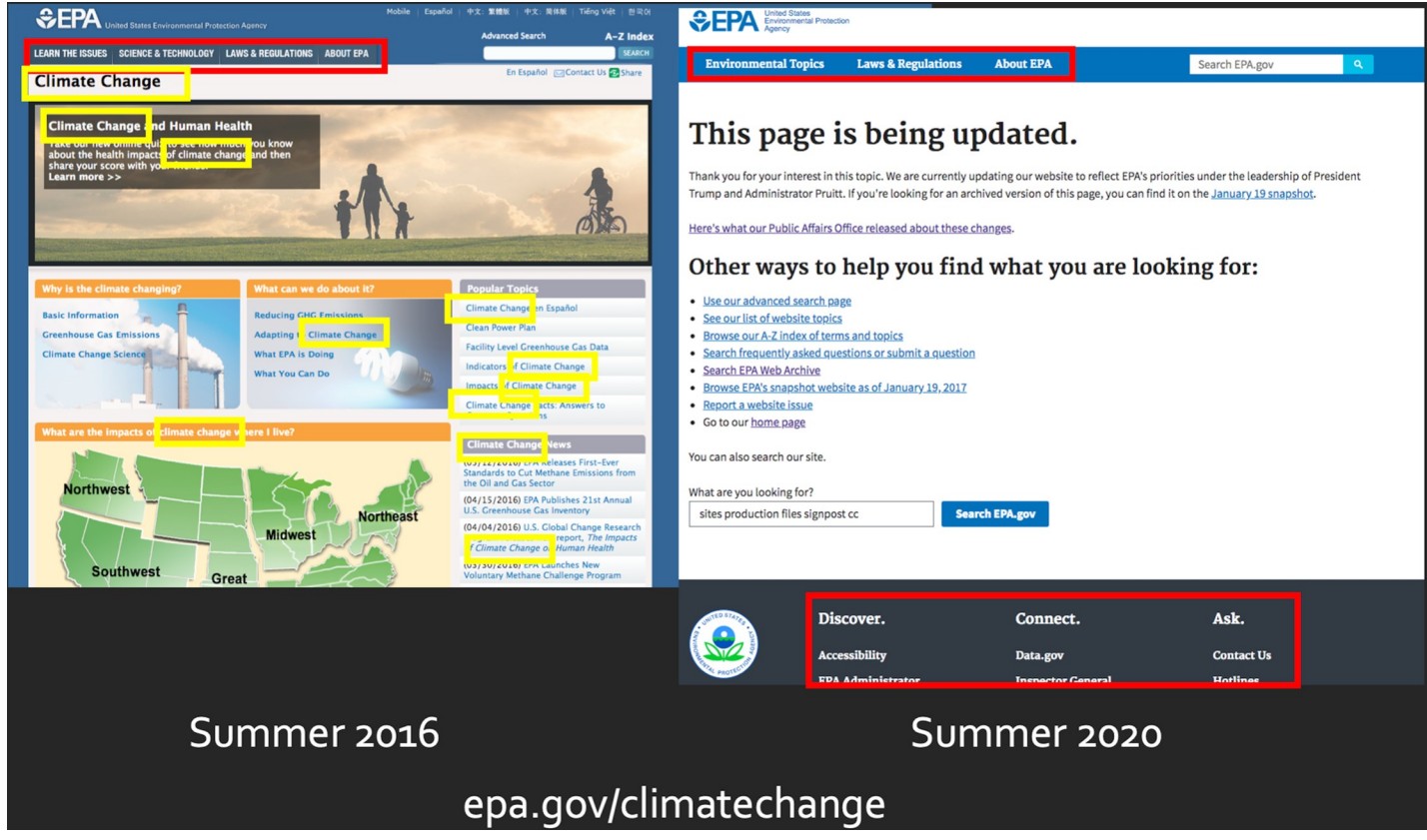

**Fig 1. An example of how we scraped webpages for our term analysis.** We excluded information in navigation menus and in footers (red blocks) and included mentions anywhere else on the page (yellow boxes).

mention or use in a corpus of documents (e.g. [18, 80, 81]). Content analysis cannot fully capture the discursive context in which terms are used. Because ours is an attempt to examine and visualize web changes at scale, this is acceptable—we use content analysis in order to broadly illustrate changing usage of keywords, if not the full meanings associated with their use. Our previously published reports (and media coverage of them) provide this more page-specific context.

We selected every page in EDGI's database that had at least one snapshot saved in the Wayback Machine in both the first half of 2020 and the first half of 2016—a paired-page sample (Fig 1). Of the 40,378 URLs in EDGI's original list, 9,144 met this criterion and a few others, including URL length (we excluded very long URLs, which suggest a page is less important) and page type (e.g. we attempted to exclude news pages). Since we cannot go back in time to archive pages as they used to be, we chose a sample that maximizes our coverage of federal webpages pertaining to the environment.

We chose the first half of 2016 in order to avoid cases where Obama-era staff preemptively distanced themselves from climate change immediately after Trump's election, such as when EPA's Climate Ready Water Utilities renamed itself Creating Resilient Water Utilities [82]. However, there are far fewer snapshots in the Wayback Machine for the first half of 2016 than there are for the first half of 2020 because the End of Term archiving effort began only after the November election, and the archiving of pages driven by our website monitoring started in earnest in early 2017. While we cannot make claims about the entirety of the federal agency web domain, the sample size is sufficient for testing the efficacy of our methods. Our sample is

purposive—based on pages we determined early in the Trump administration would be important to follow—and opportunistic—based on pages captured by the Wayback Machine.

We scraped each of the snapshots for both 2016 and 2020, counting the use of key terms. By scraping, we mean that we used an automated program to acquire the web page content of that snapshot from the Internet Archive's Wayback Machine (see Supplemental Materials B in S1 File). The terms we chose to include in this analysis were selected empirically. We reviewed the EDGI website monitoring team's 35 published reports, two white papers, and two public comments, and identified the most frequently changed words to serve as keywords for the analyses presented in this paper: "adaptation," "air quality," "clean energy," "climate," "climate change," "energy independence," "emissions," "greenhouse gases," "hydraulic fracturing," "resilience," "sustainability," "unconventional gas," and "unconventional oil."

We also performed a network analysis (e.g. [83]), charting hyperlinks between pages on EPA's website and assessing their changing connections over time. We perform only the most basic descriptive statistics on this changing structure, including metrics such as the number of connections (edges). Our goal is to illuminate access changes at a broad level, rather than to report on the condition of EPA's website structure, such as by statistically determining components or centrality.

In our network analysis, we focused solely on EPA. We did this for both technical reasons —we do not have the computing capacity to create a network graph of connections between pages across the entire federal domain—as well as epistemological ones: EPA is the US's primary environmental rule-making body and epa.gov is one of the most visited sources for environmental information and education. Our previous work had indicated that epa.gov had been modified substantially [7, 58]. We found 2,478 EPA URLs were available from the Wayback Machine for both timeframes and we scraped these snapshots. From each snapshot's code, we extracted the links made to other EPA pages in our list. From this, we built a matrix to record connections between pages. Unlike for the content analysis, we included the navigation menus and footers on each page in our search. The use of "climate change" in a navigation menu is meaningful from an access perspective, not a content one. By not counting it in our content analysis, we tried to capture—as much as possible—the most meaningful, unique uses of the term a user would come across.

## Results

### Documenting overall changes to access

We first wanted to measure to what extent the Trump EPA had narrowed or even eliminated access to climate- and environment-related pages. To do this, we visualized the network graph of as many EPA pages as possible. In particular, we figured out a way to visualize changes to this graph (*deleted* connections especially) that incorporated directionality, or, the fact that A->B might no longer be linked, but B->A is.

A significant portion of EPA's website has been dismantled over the past three years, largely due to the removal of epa.gov/climatechange in the first 100 days of the Trump presidency. We found that epa.gov/climatechange was one of the 35 most linked-to EPA pages at the end of the Obama term because it was included in the navigation menus that pointed to key areas of the epa.gov website. However, the page was not in that ranking by Summer 2020; only six pages still linked to it. In total, we found that 22,023 connections between EPA pages were removed by the agency by Summer 2020, while a slightly greater number of connections were added (26,368). The removals represent 24.4% of all the connections we sampled during the Obama period. By 2020, 20.25% of the pages in our sample were no longer linked to by any other page.

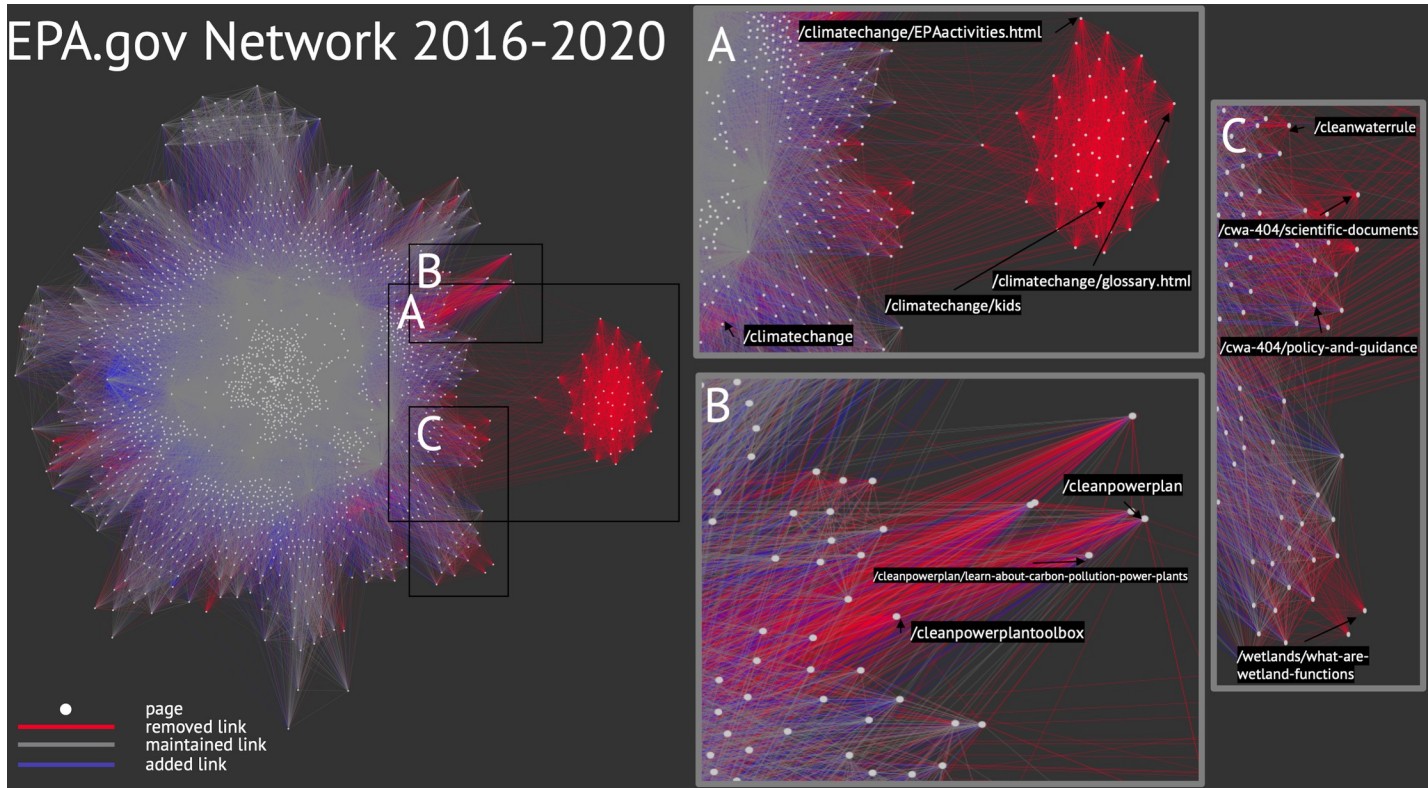

**Fig 2. Network graph of epa.gov pages.** For clarity, nodes with less than a total of 35 incoming and outgoing links are not shown.

To make these numbers stand out, we created a network graph using the Gephi visualization software (v. 0.9.2) (Fig 2). In our graph the nodes, or vertices, are individual webpages. Each line indicates a connection (called an edge) between the page and another in at least one of the timeframes. The shape of the graph is somewhat arbitrary; there are different algorithms for spacing out the nodes but these do not affect the data itself, just its visualization. The one we present here lets us see a "core" of pages that are linked to by many other pages in headers, footers, and other navigation-focused webpage elements. The pages on the outer edges of the visual are those that are linked to less or that themselves link less. The color of the line indicates whether the connection between the pages was present in both timeframes (grey), not present at Obama's end of term but added by Summer 2020 (blue), or present during the EOT but removed by Summer 2020 (red).

Fig 2 shows that most links were retained across the past four years. This reflects the relative stability of the EPA website structure for many topics and suggests that navigation among resources has not changed dramatically. We also found that while more links were added than removed, many of these additions—blue lines in the figure—go to just two pages that added dozens of links between 2016 and 2020: https://www.epa.gov/open and https://www.epa.gov/privacy. While these pages existed in 2016, they were not linked to until they were added to the menu at the bottom of each of EPA's webpages.

There are three areas in the right part of the plot indicating websites that lost many incoming and outgoing links. One of these (A) is the https://www.epa.gov/climatechange directory, which was in the main navigation menu for epa.gov in 2016. In April 2017, that directory was removed from public access. The red lines in (A) represent the many pages within the epa.gov/climatechange directory that were removed. As of summer 2020, all of the links between

climate pages and other EPA resources had been replaced by the handful of links present on the static signpost page (explaining that the content was being "updated" to reflect the Trump administration priorities). The other cluster of pages (B) is the https://www.epa.gov/cleanpowerplan site, which was removed at a similar time as epa.gov/climatechange and replaced with epa.gov/energyindependence, as the agency sought to repeal Obama's Clean Power Plan. Finally, several sets of pages (C) related to the Clean Water Act were removed in Spring 2017, as the Trump EPA began revising rules around the definition of "navigable waters" (the well-known case of the Waters of the US rule). These findings reveal the importance of web access restrictions—the Trump administration removed these pages when relevant rules were open for public comment.

## Documenting overall changes to terms

Our content analysis assessed total language changes by examining the use of key terms such as "climate change," "resilience," and "sustainability" across all 8,813 pages in the Wayback Machine sample that we were able to produce for both 2016 and 2020 (see Supplemental Materials A-C in S1 File). We experimented with how to effectively visualize this data, and in particular how to capture relative *and* absolute changes. We wanted to understand, for instance, when pages dropped half of their use of the terms (a relative change) but also to contextualize this in absolute terms (e.g. if a page went from 4 uses to 2, versus from 16 to 8). Our visualization challenges were compounded by the fact that many pages saw complete removals of key terms; we struggled with how to visualize zero.

We found that, overall, the use of "climate change" decreased by 38% between 2016 and 2020. There are several specific trends that Fig 3 made clear for us. First, the complete removal of "climate change" on many pages is suggested by the diagonal pattern in the bars in the bottom half of the chart. Indeed, many of these complete removals of "climate change" reflect the pages themselves being removed from public access, especially those within the epa.gov/climatechange directory. Second, many pages that used "climate change" five to 20 times in 2016 saw significant reductions in the term in 2020. This is made visible by the opaquer red bars in the left part of the graph. Finally, pages that started with minimal counts of "climate change" were more likely to increase their use. For instance, several pages with four uses of the term in 2016 added a handful more uses by 2020.

## Analyzing term changes against one another

We were also curious to examine the changing use of terms in relation to one another. For instance, beyond assessing the shifts in mentions of "climate change" itself, we evaluated the use of "climate change" relative to terms such as "resilience" and "sustainability" that researchers have suggested are weak but more politically-palatable substitutes that diminish our ability to conceive of and act upon the climate crisis (e.g. [84, 85]), and that we ourselves had witnessed in our weekly monitoring [7, 82]. We are not claiming that such weakening is true; we are simply using these keywords for a theoretically-informed test of how term comparison—and, ultimately, broader shifts in discourse—could be visualized.

We developed Fig 4 to assess how a discussion of "climate change" may have been replaced by a discussion of "resilience." We use "small multiples"—a series of similarly structured charts—to visualize how the 186 pages that used both "resilience" (y-axis) and "climate change" (x-axis) in 2016 used these terms in 2020 [78]. Arrows encode the change for each page. Arrows aiming "northeast" (Fig 4B) indicate pages that increased their use of both terms. Southeast-pointing arrows indicate pages where "climate change" increased and "resilience" decreased (Fig 4C). As expected, there were very few of these pages (nine). We expected many

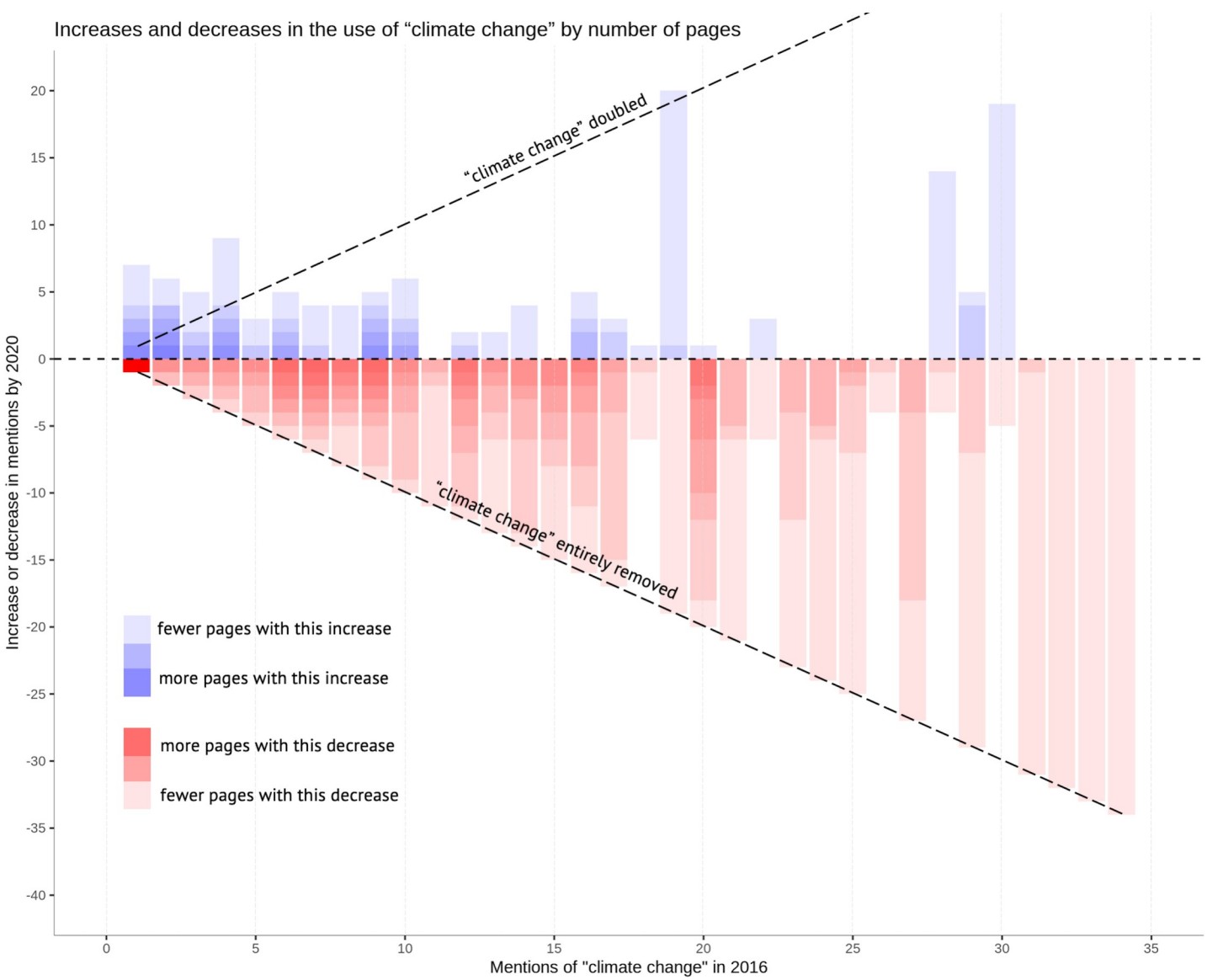

**Fig 3. Increases and decreases in the use of "climate change" across 2,085 federal agency webpages.** For example, the right-most red bar on the graph shows that there was one page that mentioned the term "climate change" 34 times in 2016 and all 34 of them were removed by 2020. The right-most blue bar on the graph shows that there was a page that had 30 mentions of "climate change" in 2016 and, by 2020, 18 more mentions of "climate change" had been added to that page. Not shown: pages with no change (n = 834) and pages with 2016 counts beyond 35.

northwest-facing arrows: pages diminishing their use of "climate change" while increasing their use of "resilience." While this would not necessarily confirm direct substitutions of "resilience" for "climate change," it would show us that, on balance, these pages shifted the emphasis from one term to another. Fig 4A shows that changes where the frequency of the term "climate change" declined and "resilience" increased occurred on 21.5% of pages we examined. For instance, EPA's Climate-Ready Water Utilities program was renamed Creating Resilient Water Utilities program. Using this visual approach to explore the shifting relative usage of terms, we also see many "southwest" arrows—89 pages that dropped both terms (Fig 4D). Many of these were epa.gov/climatechange pages that were completely removed from public access.

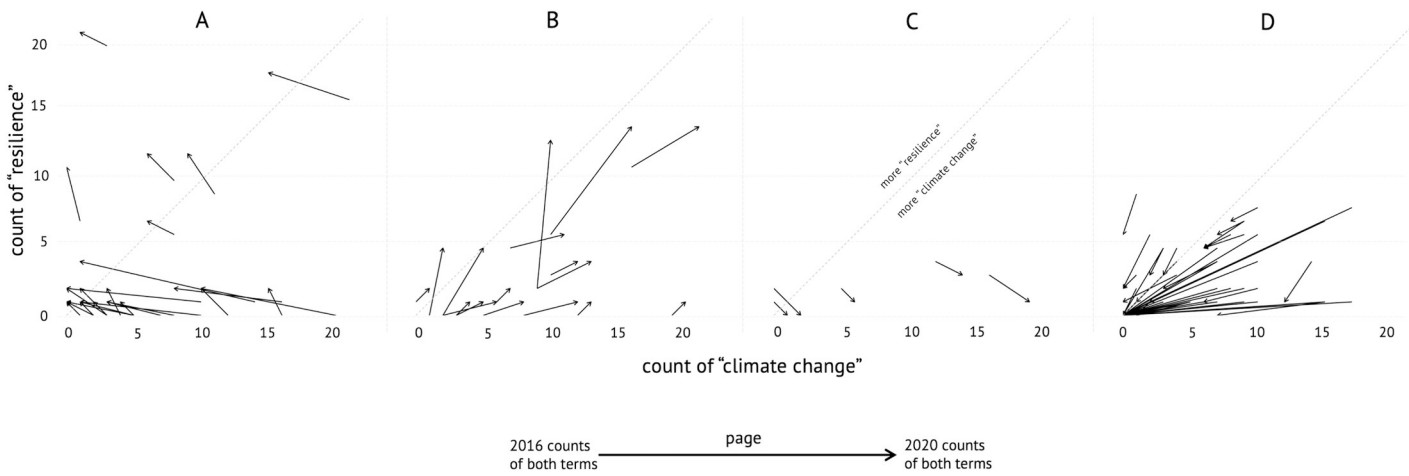

**Fig 4.** a-d. Pages that saw changes to counts of both "resilience" (y-axis) and "climate change" (x-axis). Arrows indicate the directionality of change between 2016 and 2020. (a) shows pages on which "climate change" decreased and "resilience" increased; (d) shows pages on which "climate change" and "resilience" both decreased.

What is useful about these small multiples of scatter plots is that each figure encodes absolute change—2016 counts to 2020 counts, indicated by the *length* of the arrow—while also providing a sense of relative change. The *direction* of the arrow indicates, for instance, whether "climate change" decreased while "resilience" increased, and arrows that cross the dashed line tell us about whether a page now has more uses of "resilience" than "climate change," or vice versa. The small multiples give a sense of both absolute and relative change while maximizing clarity.

## Analyzing term changes by agency

Another way to contextualize changes between 2016 and 2020 in keyword usage is to compare changes made by different kinds of federal agencies. Did agencies with closer ties to the president remove or add certain words at higher rates than others? As some journalists have argued [86], did agencies focused on research, such as NIEHS, modify keywords differently from agencies with more of a regulatory focus, like the EPA?

To analyze term usage by agency, we calculated both per page use and average per page changes. First, we summed each agency's use of "climate change" in 2020 and divided it by the total number of the agency's pages in our sample, then did the same for 2020 and plotted these against each other (Fig 5). This per page use metric provides a sense of how likely a user would be to come across "climate change" while navigating an agency's website. We then calculated the average per page change for each agency and keyword by subtracting the term's total mentions in 2016 from its mentions in 2020, and dividing by the number of agency pages it originally appeared on (Fig 6). This metric directly encodes change over time and approximates the scope of changes a website visitor might experience—if they went to a page using "climate" in 2016 and then returned in 2020, how different would their experience be? We determined that this was the most effective measure of how agencies altered their usage of keywords because it is a single value that incorporates pages with substantial removals or additions, as well as those where a term was not altered. Other single measures are less effective. For instance, the median value of per page changes would fail to reflect pages with dramatic changes in the use of a term.

Fig 5 depicts the per page use, by agency, of "climate change." Much like the small multiples in Fig 4 above, the x-axis of this chart represents 2016 counts, while the y-axis represents

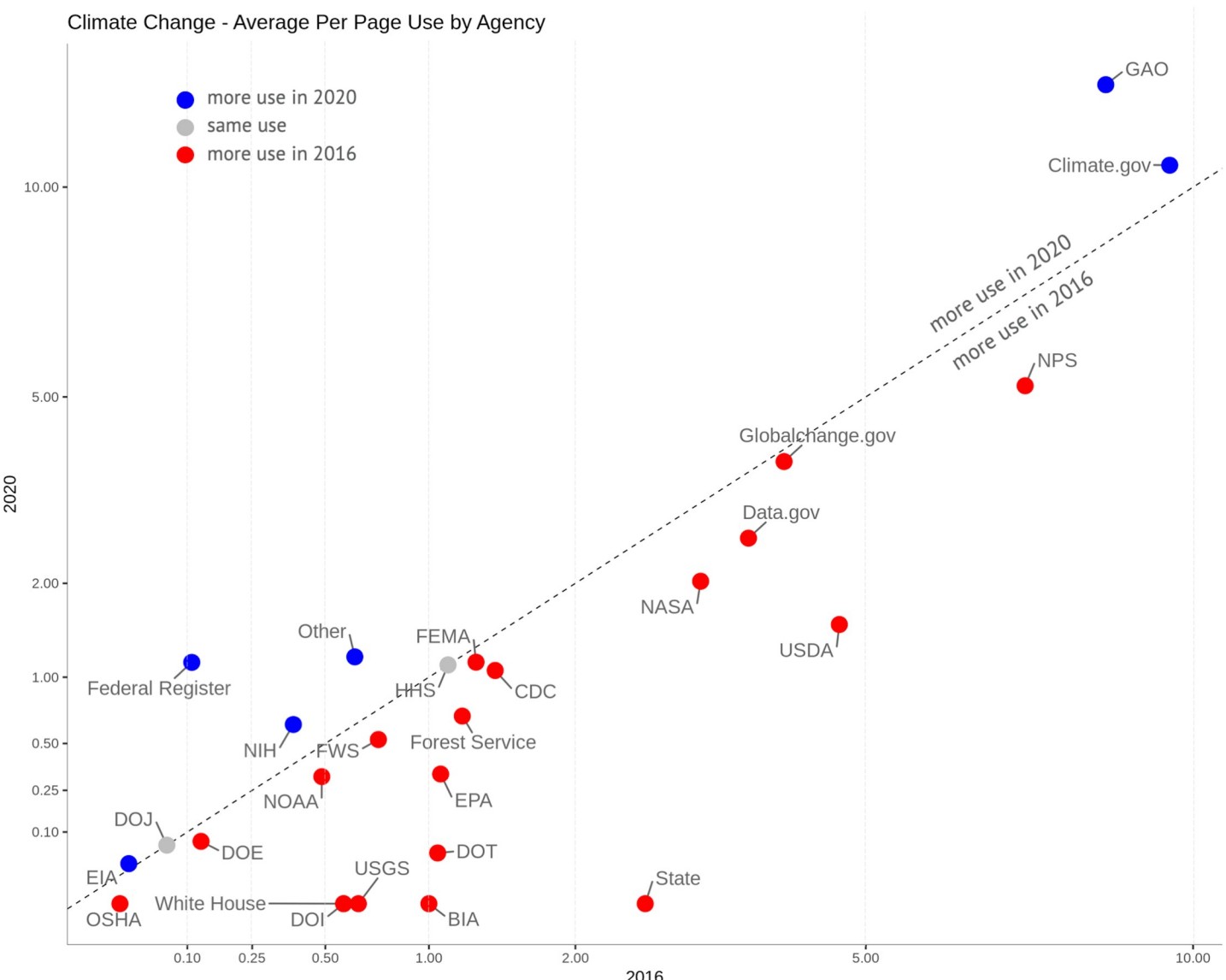

**Fig 5. The per page use of the term "climate change" for each agency in 2016 and 2020.** The trend line represents no change (x = y). The x and y axes are square root scaled. Not shown: agencies that never used the term. See Table 1 for acronyms.

counts in 2020. Thus, change is encoded in the position of the dot. Agencies below the line, in red, decreased their per page use of "climate change." Most agencies fall into this category, though some—such as the White House and the State Department—tended to remove mentions of "climate change" more than others, such as NOAA, which made only a slight decrease in its per page use of the term. Those in gray made no change, but agencies in blue such as the National Institutes of Health (NIH)—as well as the Federal Register (FR) and the Government Accountability Office (GAO) website—actually increased their use. The FR publishes proposed rules, including proposed repeals of rules. Between 2016 and 2020, the Trump administration's rollbacks of Obama-era climate regulations were all published in the FR, and necessarily referenced the original language of the rule. The GAO fulfills requests from Congress, especially calls for investigations. The increase in usage of "climate change" on the GAO website

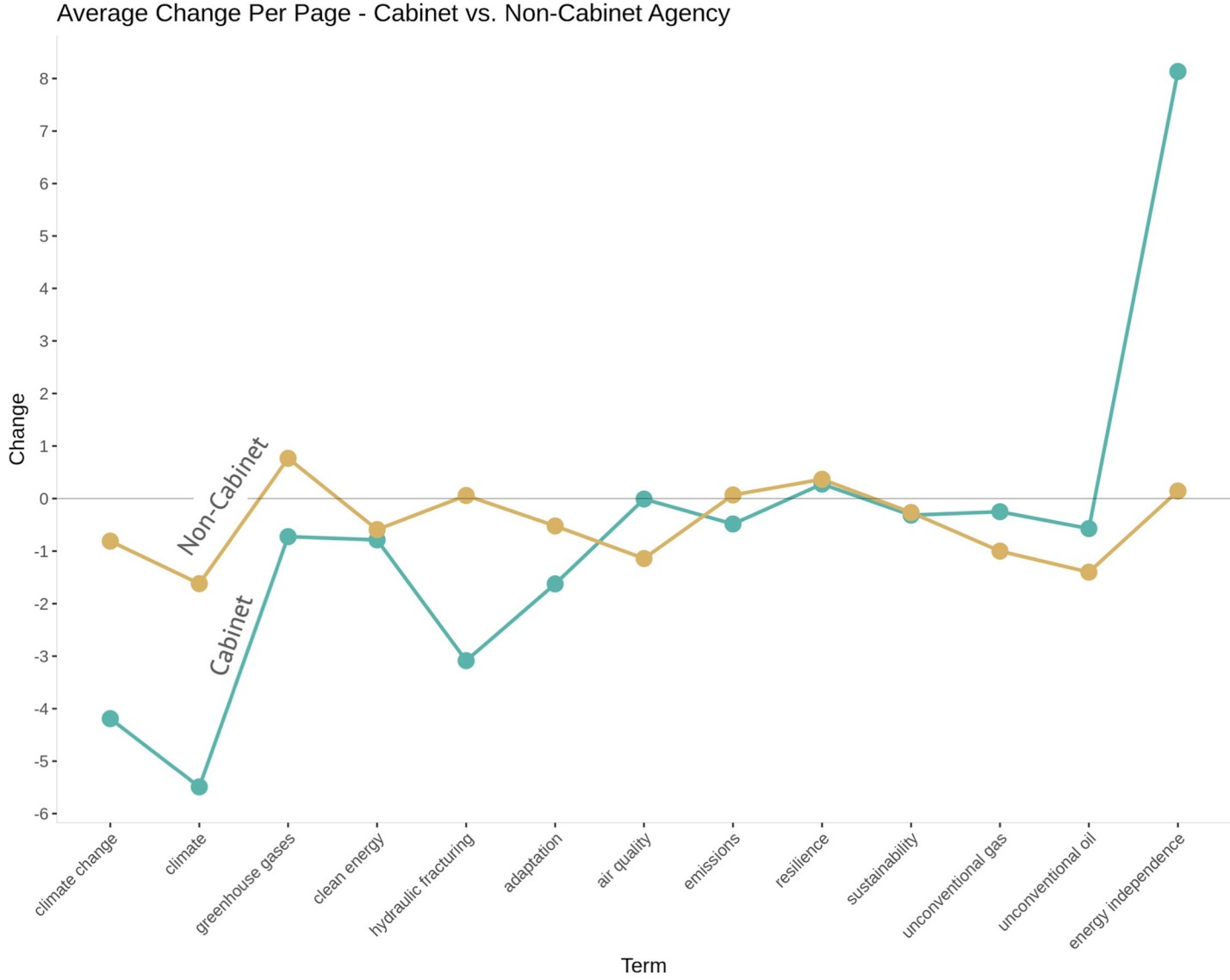

**Fig 6. Average change per page on Cabinet vs non-Cabinet agency websites.** Terms are ordered left to right by our empirically driven expectations of decreases or increases in usage.

represents an increase in congressional questions about agencies' actions around climate change. The only executive branch agencies to increase their per page use of the term "climate change" between 2016 and 2020 were the NIH, Energy Information Administration (EIA, an independent agency within the Department of Energy), and Climate.gov (a website provided by NOAA to increase public understanding of climate change).

To further characterize these differences in term usage between agencies, we assigned each agency as having a regulatory function or not, having a research mission or not, and being of Cabinet level or not (see Table 1), and made composite samples on that basis. These assignments were made based on thematic coding of each agencies' mission statement. While results show some differences between regulatory and non-regulatory agencies, and between agencies with a research mission and those without, the clearest, most consistent signal was between Cabinet-level versus non-Cabinet agencies.

Members of the Cabinet (except for the vice president) are appointed by the president with Senate approval. Once confirmed, they serve at the pleasure of the president, who can dismiss them at any time without the approval of the Senate. In addition to serving as administrators of their appointed offices, Cabinet members advise the president on diverse topics of national importance. The president's Cabinet officially includes the heads of federal departments and in practice includes administrators of a few other federal agencies as well. The EPA, while not officially a Cabinet agency, is usually offered a Cabinet-level position, and the EPA Administrator is included in the Trump Administration's list of Cabinet members [87].

To aid our visualization of agency differences, we ordered terms on the x-axis by the relative degree of removals or additions we predicted based on our previous weekly monitoring and published reports. We had documented myriad removals of the term "climate change," and expected its frequency of use to have dropped the most in our dataset (e.g. [82]). Thus, "climate change" is placed in the left-most position on the x-axis. We had documented some additions and removals of other terms, such as "adaptation," so we placed them in the middle of the x-axis (e.g. [88]). We had documented additions of "sustainability" and an entire subdomain (epa.gov/cleanpowerplan) redirected to an "energy independence" page, so these terms are placed at the right end of the x-axis. Our sample—drawn from pages present in both 2016 and 2020—largely underestimates additions of the terms "unconventional oil," "unconventional gas," and "energy independence," which are frequent on *new* pages created by the Trump administration, such as EPA's Unconventional Oil and Gas website, www.epa.gov/uog.

Visualizing the average per page change reveals that Cabinet and non-Cabinet agencies vary in how they altered climate-related keywords (Fig 6). For Cabinet agencies, there is a notable reduction in terms such as "climate," "climate change," "greenhouse gases," and "clean energy," and an increase in the use of a highly politicized term on the other end of the spectrum, "energy independence." In between, there are decreases in a few terms, such as "adaptation," where we had observed both removals and additions.

Non-Cabinet agency websites show a less pronounced pattern of climate-related language changes. These average per page changes are closer to zero for all terms, and do not follow the decrease-to-increase pattern we would expect based on our weekly website monitoring. The terms with the most increased use in non-Cabinet agencies are the vaguer terms "resilience" and "emissions" as well as "greenhouse gases." Besides "climate," the term with one of the steepest decreases in use in non-Cabinet agencies is "air quality," which we observed as changing very little on Cabinet agency websites.

One of the most dramatic differences between Cabinet and non-Cabinet agencies clarified by Fig 6 relates to the term "hydraulic fracturing." The decrease in usage on Cabinet agency webpages is largely due to the removal of the term from an internal list of links to resources developed in "EPA's Study of Hydraulic Fracturing and Its Potential Impact on Drinking Water Resources" page—renaming links from "Hydraulic Fracturing Study Fact Sheets" to "Fact Sheets," for example. This is a real change in the number of times a visitor would see the term hydraulic fracturing when visiting the EPA website, but it is not necessarily a substantively important one. Changes such as these underscore why ground-truthing calculated changes remains important, as does recording the absolute and average page change numbers.

We followed up on inter-agency differences in per page term usage by creating a dot plot (Fig 7). The figure usefully illustrates the range and distribution of changes for both Cabinet and Non-Cabinet agencies. The size of each dot represents how many pages saw that degree of change in keyword usage and the color reflects agency status (Cabinet = tan; non-Cabinet = green). In general, Cabinet agencies have larger ranges and more decreases in the use of terms than non-Cabinet agencies. The largest ranges are for "climate change" and "climate," and there are more extreme decreases for those terms than there are increases. The

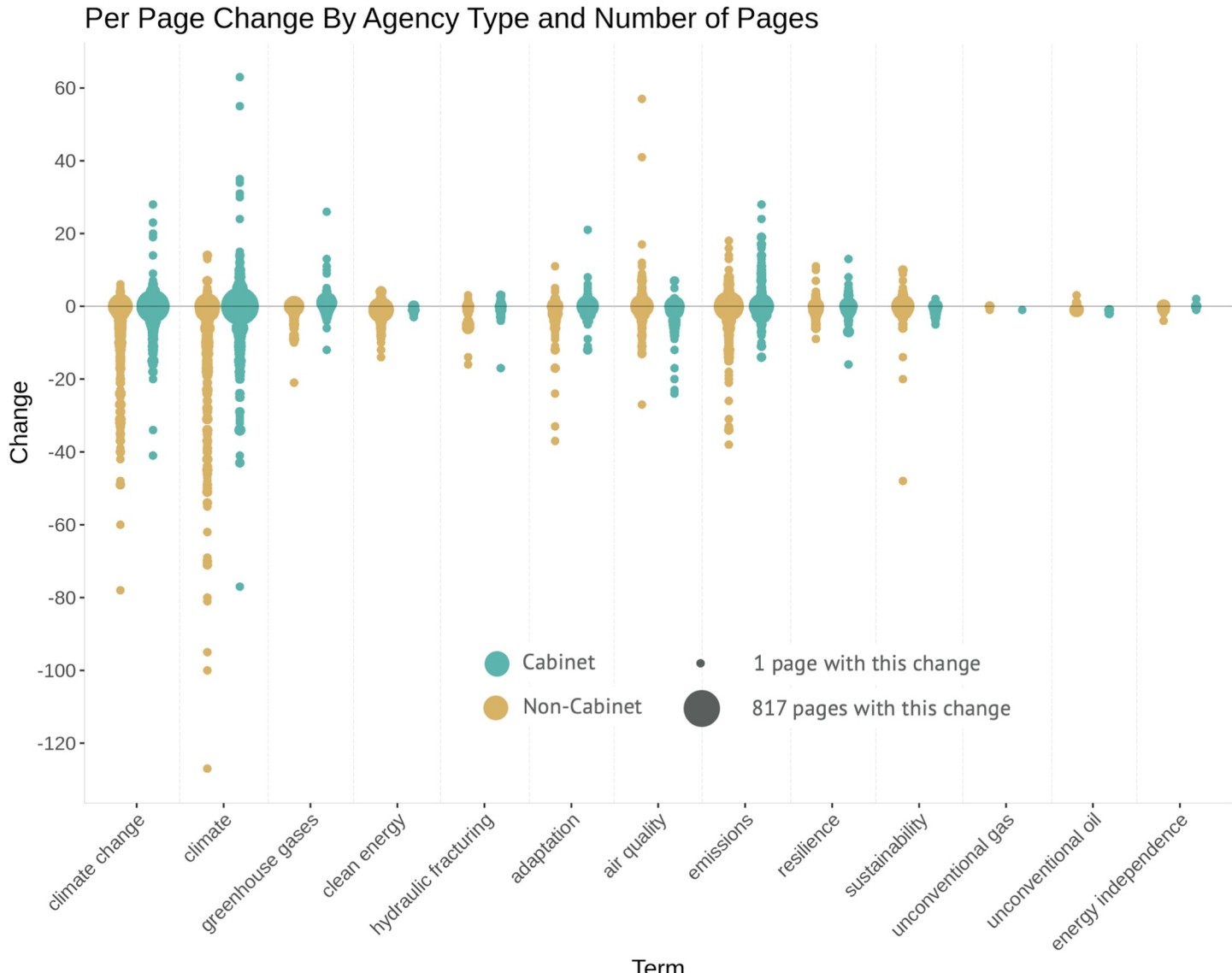

**Fig 7. Distribution of changes to climate-related terms on Cabinet and non-Cabinet agency websites.** Terms are ordered left to right by our empirically-driven expectations of decreases or increases in usage.

terms "adaptation," "air quality," and "emissions" have wide ranges as well, especially in Cabinet agencies where they replaced more specific climate-related language (such as "emissions" replacing "greenhouse gases" [89]), and decreases were driven by the removal of EPA's Climate Change website. Terms with very small ranges of change typically had fewer pages that included those terms. For example, only 52 of the pages in our sample included the term "unconventional oil," so its narrower range is in part an artifact of the sampling.

## Analyzing term changes by page visibility

One final way to analyze shifts in language on agencies' websites is to look at *where* keyword changes occur within website domains. We compared changes on pages "higher" in a domain's hierarchy versus those "deeper." We deemed URLs with two subdirectories or fewer as

"higher" in the agency's web infrastructure, more visible to a browsing public. URLs with three or more subdirectories are "deeper," less visible, and often centered on more specific subtopics. For example:

https://www.epa.gov/climate-researchwhich has just one / after epa.gov, is at a higher level than:

http://www.usda.gov/oce/climate_change/adaptation/adaptation_plan.htmwhich has four slashes after usda.gov. Related, the EPA "/climate-research" page covers a more general topic and is generally easier to navigate to from elsewhere in the EPA website. The USDA ". . ./adaptation_plan" page covers a specific subject matter, takes more navigation to discover, and is more "in the weeds" than a central landing page for information.

We found that the term "climate change" decreased more on lower level, deeper pages than on higher ones, and terms like "resilience" increased more on higher-level, highly visible pages than on lower level ones (Fig 8).

We also broke these page level changes down by agency (Fig 9). This combination of page level-based charts with interagency-based plots provide opportunities to compare and contrast how different kinds of agencies made terminology changes. While EPA dramatically reduced the use of "climate change" on deeper pages, USDA, USFS, and DOE removed the term much more from higher-level URLs.

This technique helped us identify patterns consistent with other actions by the Trump administration to influence public perception and undermine trust in government and science by controlling the narrative, even when it runs contrary to verifiable facts (e.g. the size of his inauguration crowd or the direction of Hurricane Dorian in 2019). The public is more likely to arrive on pages higher in a website's URL structure both by searching for broad topics (e.g. climate change, hydraulic fracturing, natural gas) and by navigating within an agency's website. Thus, the most efficient way for an administration to adjust public perception of an issue would be to alter these more visible pages. Edits to climate-related terms on such pages, targeting the most visible discussions of climate change issues, are consistent with the climate crisis's politicization. While we did not include news releases in our analyses, this approach was taken by the USDA. Climate change research continued to be conducted by the agency throughout the Trump administration and could be found on deep webpages, but it was censored from news releases intended for public consumption [90]. Meanwhile, keywords such as "resilience" that appeared to be more palatable to the Trump administration saw increased use on highly visible pages.

Another way for administrations to shape public understanding of an issue would be to more thoroughly remove the resources addressing it. That changes to climate-related terms often occurred on deeper pages likely reflects how entire swaths of federal webpages were removed from public access [56]. The EPA took this approach repeatedly. In addition to removing its Climate Change website, the EPA also removed whole websites related to regulations it rescinded under the Trump administration, such as those for the Clean Power Plan and the Clean Water Rule. The larger decrease in climate-related terms on less visible EPA pages is indeed driven by these removals.

Changes to climate-related terms on deeper pages may also represent a more thorough masking of information. For example, career staff may attempt to preserve substantive material by casting it in politically neutral language or fundamentally reconceptualizing it. Researchers have found that during the Trump administration federal agency personnel engaged in self-censorship, particularly at the EPA and DOI, and particularly for climate change research. In a 2018 survey, 31% and 26% of respondents at the EPA and DOI, respectively, agreed or strongly agreed that they avoided working on climate change or using the term "climate change" in relation to their work [56].

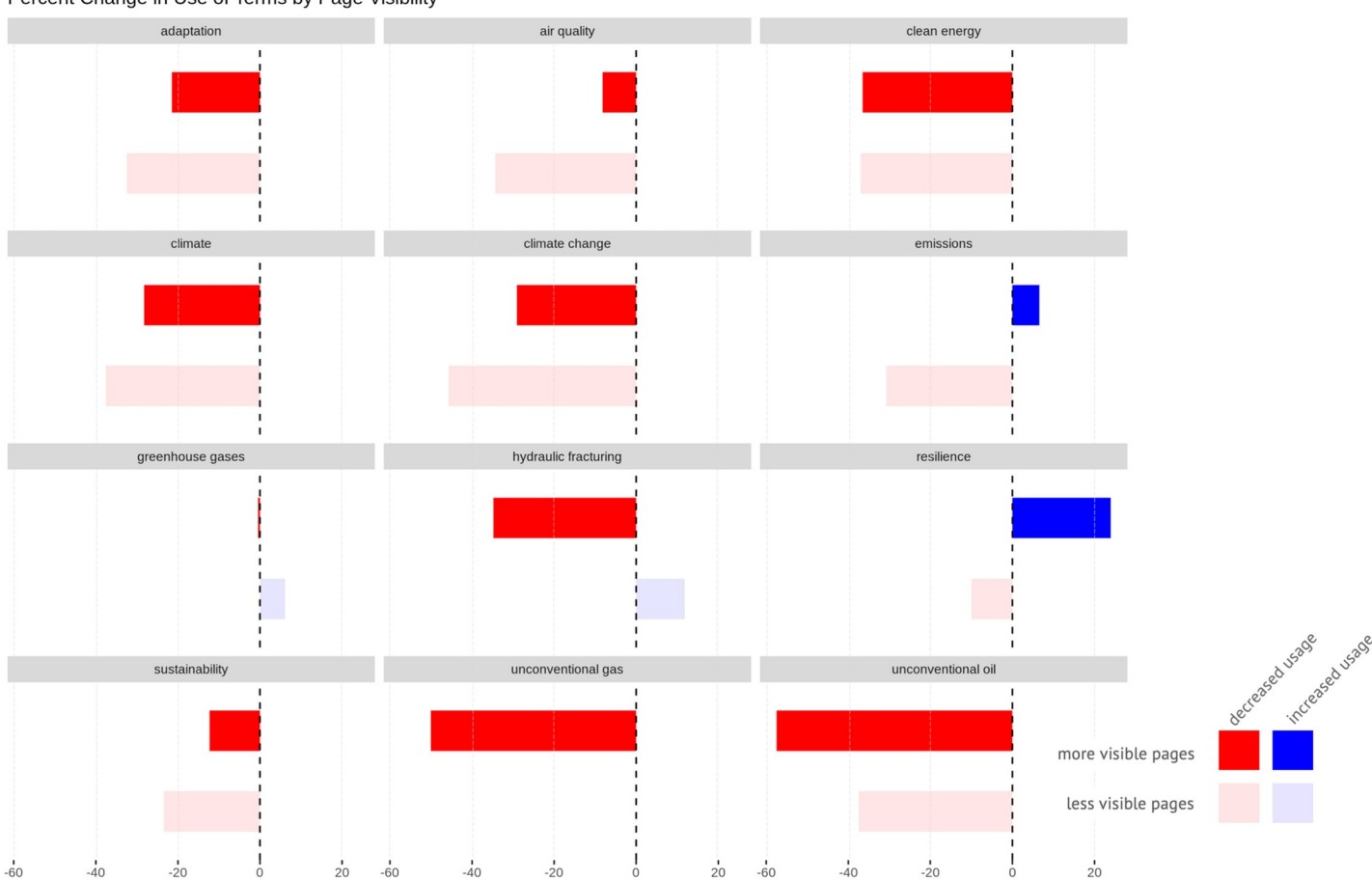

**Fig 8. Percent change in the use of key terms by page visibility.** More opaque bars represent more visible pages; more transparent bars represent less visible ones. Blue indicates additions; red represents removals.

## Discussion

Evaluations of government websites and their role in public communication often are agency- or website-specific and involve in-depth case studies. Methods that are automated, quantitative, and visually interpretable can complement this research at the scale of the federal government's web presence. Working at this scale is important as the public's use of digital media for information-seeking intensifies. Working on the timeframe of a presidential term is also of critical importance to assess prior administrations' web governance policies and offer a blueprint for incoming administrations to restore and improve the accessibility of public information. Here, we discuss how well each data visualization supports holding the government accountable for its stewardship and dissemination of publicly-funded environmental information on the web.

Existing research has documented government website management and theorized how access to web resources shapes public engagement with issues (e.g. [34]), but it has not often done so to scale or in a visual way. To attempt this, we visualized the EPA's network graph. While this technique does not allow us to easily show access changes for any specific page, it does allow us to see the impacts resulting from the removal of hub pages. For example, we had documented the EPA's removal of its climate change pages prior to creating the graph, but it

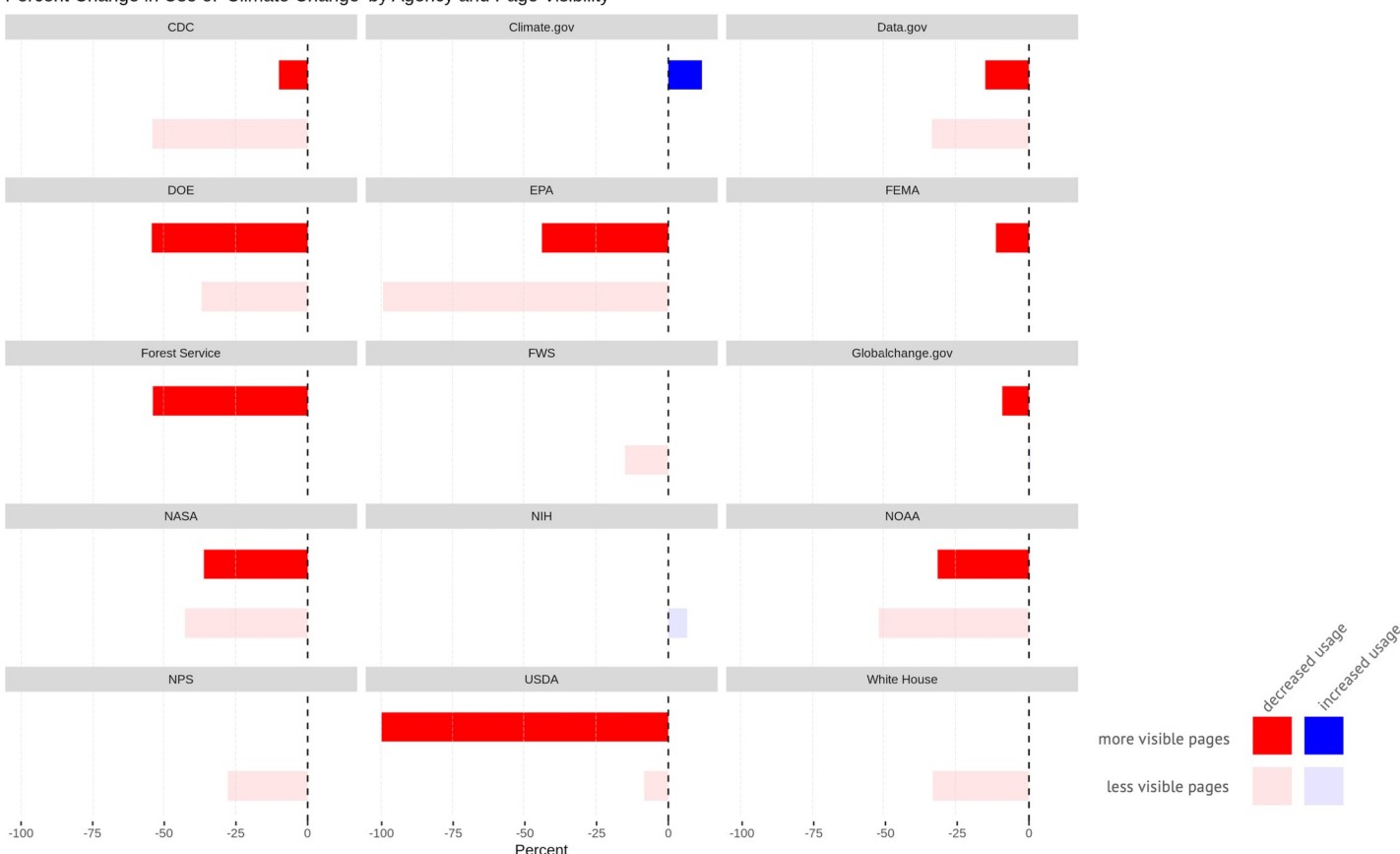

**Fig 9. Percent change in the use of "climate change" by agency and page visibility.** Not included: pages where "climate change" was not mentioned during the Obama era. Not shown: agencies with fewer than 15 pages where "climate change" was mentioned, as well as the "Other" category of agencies.

effectively visualizes the scope of access restrictions. Visualizing network graphs can address fundamental questions about a website's structure, such as how the interlinking of pages changes over time or if certain resources are becoming more difficult to navigate to or find.

Existing research also suggests that the content on websites may shape public understanding and engagement (e.g. [12, 34]). Our paired-page sample approach allows for a rapid determination of keyword changes that would be indicative of this. We illustrated shifts in terminology using various visualization techniques. Term counts can be plotted to show both absolute and relative changes, and by page or in meaningful aggregate categories, such as by agency. Plotting two terms—as we did in Fig 4—helped us to better understand the flavor of keyword changes—whether it was in favor of one term over the other. A benefit of this approach is its applicability beyond the environmental realm to anywhere where usage of keywords matters, such as the important humanitarian shift from "illegal" to "undocumented" immigration, or the shift from "domestic" violence to "intimate partner" violence. This dynamic may be especially relevant when considering people with limited knowledge of a subject, for whom the most common keywords may be the only ones that are initially recognizable. As with all our techniques, this one is most informative alongside spot-checking that provides context regarding why an alteration may have occurred.

We sought to further characterize and not just document these changes, first by inspecting patterns in them between types of agencies and across types of keywords. In our Cabinet/non-

Cabinet visuals (Figs 7 and 8), terms were ordered by the number of times they appeared in our own previously-published reports, but they could be ordered along the x-axis in any way based on researcher expectations. This empirically-driven approach provides a point of comparison between researchers' qualitative experience or case studies and large-scale data. In our analysis of federal environmental agency websites, we expected that the terms for which we documented removals in our reports would be those whose usage decreased the most across the dataset, and those for which we documented additions would have the most increased usage. When analyzing the average term usage change per page, this assumption held true for Cabinet-level agencies but not for other agencies that were either independent or smaller agencies under the umbrella of a cabinet agency. In other words, we found that the most politicized agencies—Cabinet-level ones—made the most changes and to terms that have polarizing connotations.

Since our goal in relation to previous research was to prototype scalable, rapid techniques for assessing and visualizing web changes, we sought as straightforward an approach as possible. Analytical methods involving Natural Language Processing could be applied as next steps. We found that the average per page term change offers a compelling perspective on discursive shifts, especially in comparing between agencies. It provides the single most meaningful number to quickly assess a change as positive or negative and with some measure of extremity, though it does not capture variation in the actual usage of the terms on different pages. For most terms, many pages included no change in usage. For certain terms, a few pages reflected extreme usage changes, including a single page on the EPA website with 78 fewer uses of "climate change" in 2016 than in 2020. To visualize variation within the agency groupings, we created a dot plot of term usage changes on individual pages (Fig 8).

We found page-level analysis to be a powerful way of understanding the effects of keyword changes. We visually identified patterns where some sets of pages—either more visible ones or lower-level, content-specific ones—witnessed more changes. Changes made to more visible landing pages could occur for a number of reasons but are consistent with efforts to alter public perception of the issues the pages address, as they are the ones the public is most likely to encounter. Changes made to deeper, more specific subject matter pages are consistent with efforts to alter public perception if they are part of thorough purging of information, as occurred with the removal of the entire epa.gov/climatechange directory [91]. Changes made to lower-level pages are also consistent with self-censorship and agency staff desires to have programs "fly under the radar" [54, 56]. Identifying patterns of where in the web infrastructure changes occur could provide a basis for further investigation to understand the context in which changes have been made, and to hold the government accountable for those changes.

Existing research in e-government provides case studies that can shed light on possible causal factors for why website changes occur [32]. That has not been our own goal here; we have not sought to provide an authoritative, mechanistic, or causal explanation for why changes were made to federal environmental webpages. The most basic explanation is simply that they are allowed to occur; a lack of rigorous digital archival policy in the US means we should reasonably anticipate them. In addition, these websites are expected to change to some degree, administration to administration. Our analyses could lead to Freedom of Information Act requests to more fully substantiate causal explanations.

## Conclusions

Relatively little research has sought to comprehensively characterize changes to US federal government websites over the past four years. More generally, we need to further our understanding of how these websites change under different administrations and what standards they are

held to in managing their online content. This would benefit assessments of whether changes to information are justifiable, who is responsible for them, and what their present and future impacts may be. Researchers have shown government agency websites are important means by which issues are framed for the public (e.g. [34]) and by which the public can assess the operation of government. Websites therefore do not simply reflect politics but mediate it, and changes to them should be characterized for the sake of evaluating harms to public understanding and democratic participation.

Responding to this, we drew from content and network analysis and implemented five different data visualization techniques to illustrate changes to government websites between 2016 and 2020: we (1) documented overall changes to access with a network graph, (2) documented overall changes to terms with bar charts, (3) analyzed terms against one another using small multiples, (4) analyzed term use changes by agency with dot plots, and (5) analyzed term use changes by page-level with bar charts. We suggest that these are techniques that can be used to hold federal agencies accountable for their digital presence and for their science and environmental policies.

Findings from our case study are best summarized in four main points. First, we found that usage of "climate change" and related terms significantly decreased, and access to many pages describing these terms was restricted. Second, our page level analysis reveals these decreases in usage of the term "climate change" occurred more frequently on lower-level pages relative to more visible ones, consistent with an effort to alter the public perception of climate science through a thorough purging of information. Third, we demonstrated that visual techniques can help us to see web changes at scope and scale. Lastly, throughout our analysis we sought to emphasize the current permissibility and legality of changes made to US federal websites. In other words, in the absence of a legal framework to protect, preserve, and ensure access to publicly-funded digital information, web content may be altered, removed, or otherwise hidden without substantive repercussions.

Ours is just one attempt to make sense of the Trump administration's changes to environmental agencies' web resources, and to suggest relatively straightforward, accountability-focused means for characterizing and visualizing these changes. Our findings underscore the need for additional research, especially text and topic analyses that get closer to the meaning of and not just counts of terms. Likewise, our widely-scoped content analysis can point out specific pages, sites, and agencies where researchers might focus detailed qualitative case studies to better explain why changes occurred. In addition, there is opportunity to conduct more sophisticated network analyses of federal environmental agencies beyond the EPA, in order to better characterize access to content across federal websites.

Our work also leads us to propose a variety of recommendations for web governance in the public interest. After all, federal websites are publicly-funded resources, raising fundamental questions about whose right it is to alter and remove them. Following Lamdan, we call for legally-enforceable policies to ensure that public digital information hosted on federal websites is protected [62]. While website information should indeed be updated–for instance, to inform the public of evolving scientific understandings, emergency situations, or policy updates—a historical record of the page should be preserved. Our efforts relied on a public though non-authoritative source, the Internet Archive and its Wayback Machine. We suggest that the federal government's responsibility towards its websites should include the institutionalization of periodic server crawls to ensure that agencies are on the record for the changes they implement. But because preservation is not the same as accessibility, legal frameworks for archiving must also include provisions that ensure public access to the archive. When access to public digital information—including historical information—is reduced, the ability to effectively contribute to democracy and to make informed decisions is curtailed.

## Supporting information

**S1 File.**
(DOCX)

## Acknowledgments

This work would not have been possible without the Internet Archive, a non-profit organization that provides the most comprehensive archive of federal webpages in the United States. In lieu of a federally mandated archive, this independent service makes the work of website monitoring possible.

We would like to thank Phil Brown, Chris Sellers, and Ed Summers for their comments on the manuscript. In addition, we are indebted to former and current contributors to and leaders of EDGI's website monitoring team, including: Dan Allan, Anastasia Aizman, Cole Alder, Chris Amoss, Maya Anjur-Dietrich, Andrew Bergman, Rob Brackett, Steven Braun, Kelsey Breseman, Madelaine Britt, Ed Byrne, Jesse Card, Raymond Cha, Janak Chadha, Morgan Currie, Justin Derry, Jon Gobeil, Bryan Gygi, Pamela Jao, Sara Johns, Abby Klionsky, Stephanie Knutson, Katherine Kulik, Rebecca Lave, Michelle Murphy, Kevin Nguyen, Kendra Ouellette, Alejandro Paz, Lindsay Poirier, Matt Price, Toly Rinberg, Sara Rubinow, Justin Schell, Lauren Scott, Nick Shapiro, Miranda Sinnott-Armstrong, Chris Tirrell, Lizz Ultee, Julia Upfal, Tyler Wedrosky, Adam Wizon, and Jacob Wylie.

## Author Contributions

**Conceptualization:** Eric Nost, Gretchen Gehrke, Grace Poudrier, Marcy Beck.

**Data curation:** Eric Nost.

**Formal analysis:** Eric Nost, Gretchen Gehrke.

**Investigation:** Eric Nost, Gretchen Gehrke, Grace Poudrier.

**Methodology:** Eric Nost, Gretchen Gehrke, Marcy Beck.

**Project administration:** Eric Nost.

**Resources:** Eric Nost.

**Software:** Eric Nost.

**Supervision:** Eric Nost.

**Validation:** Eric Nost, Gretchen Gehrke, Marcy Beck.

**Visualization:** Eric Nost, Gretchen Gehrke.

**Writing – original draft:** Eric Nost, Gretchen Gehrke, Grace Poudrier, Aaron Lemelin.

**Writing – review & editing:** Eric Nost, Marcy Beck, Sara Wylie.

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
