## [Decision Letter · Decision Letter 0]

29 Oct 2020

PONE-D-20-23946

Explaining change and fostering accountability in digital environmental governance

PLOS ONE

Dear Dr. Nost,

Thank you for submitting your manuscript to PLOS ONE. After careful consideration, we feel that it has merit but does not fully meet PLOS ONE’s publication criteria as it currently stands. Therefore, we invite you to submit a revised version of the manuscript that addresses the points raised during the review process.

The reviewers both provide constructive and thorough reviews of the manuscript. Prior to acceptance, the authors must address the significant concerns raised by the reviewers. In particular, R2 notes the need to strengthen the conceptual underpinning of the paper and better link the data analysis to theory. R2 also raises important methodological points that should be considered. R1 and R2 both note interpretive leaps from the data presented in the paper that may not be warranted or should be better justified before the paper can be accepted for publication.

---

## [Author Response · Author response to Decision Letter 0]

7 Jan 2021

Please see attached "response to reviewers"

---

## [Decision Letter · Decision Letter 1]

20 Jan 2021

Visualizing Changes to US Federal Environmental Agency Websites, 2016-2020

PONE-D-20-23946R1

Dear Dr. Nost,

We’re pleased to inform you that your manuscript has been judged scientifically suitable for publication and will be formally accepted for publication once it meets all outstanding technical requirements.

Kind regards,

Jacob Freeman

Academic Editor

PLOS ONE

Additional Editor Comments (optional):

Thank you for submitting you paper to Plos One.

Reviewers' comments:

Reviewer's Responses to Questions

**Comments to the Author**

1. If the authors have adequately addressed your comments raised in a previous round of review and you feel that this manuscript is now acceptable for publication, you may indicate that here to bypass the “Comments to the Author” section, enter your conflict of interest statement in the “Confidential to Editor” section, and submit your "Accept" recommendation.

Reviewer #1: All comments have been addressed

2. Is the manuscript technically sound, and do the data support the conclusions?

Reviewer #1: (No Response)

3. Has the statistical analysis been performed appropriately and rigorously? 

Reviewer #1: (No Response)

4. Have the authors made all data underlying the findings in their manuscript fully available?

Reviewer #1: (No Response)

5. Is the manuscript presented in an intelligible fashion and written in standard English?

Reviewer #1: (No Response)

6. Review Comments to the Author

Reviewer #1: (No Response)

7. PLOS authors have the option to publish the peer review history of their article (what does this mean?). If published, this will include your full peer review and any attached files.

Reviewer #1: No

---

## [Editor Report · Acceptance letter]

5 Feb 2021

PONE-D-20-23946R1 

Visualizing Changes to US Federal Environmental Agency Websites, 2016-2020 

Dear Dr. Nost:

I'm pleased to inform you that your manuscript has been deemed suitable for publication in PLOS ONE. Congratulations! Your manuscript is now with our production department. 

Kind regards, 

on behalf of

Dr. Jacob Freeman 

Academic Editor

PLOS ONE